# EVENT-AIDED DENSE AND CONTINUOUS POINT TRACKING

## ABSTRACT

Recent point tracking methods have made great strides in recovering the trajectories of any point (especially key points) in long video sequences associated with large motions. However, the spatial and temporal granularity of point trajectories remains constrained by limited motion estimation accuracy and video frame rate. Leveraging the high temporal resolution motion sensitivity of event cameras, we introduce event data for the first time to recover spatially dense and temporally continuous trajectories of any point at any time. Specifically, we define the dense and continuous point trajectory representation as estimating multiple control points of curves for each pixel and model the movement of sparse events triggered along continuous point trajectories. Building on this, we propose a novel multi-frame iterative streaming framework that first estimates local inter-frame motion representations from two consecutive frames and inter-frame events, then aggregates them into a global long-term motion representation to utilize input video and event data with an arbitrary number of frames. Extensive experiments on simulated and real-world data demonstrate the significant improvement of our framework over state-of-the-art methods and the crucial role of introducing events for modeling continuous point trajectories.

## 1 INTRODUCTION

Estimating fine-grained motion from input videos is a crucial task in computer vision with widespread applications in downstream tasks such as video compression (Agustsson et al., 2020), video frame interpolation (Xu et al., 2019; Jin et al., 2023), motion segmentation (Bielski & Favaro, 2022; Meunier et al., 2023), and dynamic scene reconstruction (Guo et al., 2023). However, most early studies are based on two-frame optical flow (Ilg et al., 2017; Teed & Deng, 2020). Although these flow-based methods can model the spatially dense motion of each pixel within adjacent frames, they suffer from the challenge of capturing the long-term dynamic changes across video sequences. With the proposal of the tracking any point (TAP) task (Doersch et al., 2022; Harley et al., 2022), using sparse points as query indexes to estimate pointwise long-term motions also draws attention. Despite significant progress, these point tracking methods are limited by data acquisition and motion modeling when dealing with complex dynamics. The sparse and independent representation of pointwise motion remains inherently incompatible with the spatially dense representation of video, and the temporally frame rate of input video is constrained by conventional shutter cameras. Consequently, accurate estimation of fine-grained spatio-temporally dense and continuous motion remains a challenging and worthwhile research problem.

The event camera is a new bio-inspired vision sensor (Gallego et al., 2022). Unlike traditional shutter cameras that expose the entire image at fixed frame rates, each pixel in an event camera independently and asynchronously detects brightness changes at the microsecond level. This unique design makes event cameras inherently sensitive to motion changes in the scene, leading to their successful application in many motion-related tasks such as optical flow estimation (Zhu et al., 2019; Hagenaars et al., 2021), motion segmentation (Zhou et al., 2021; Huang et al., 2023), feature tracking (Messikommer et al., 2023), object tracking (Zhu et al., 2023), and video frame interpolation (Tulyakov et al., 2021). However, events are typically triggered only in regions with motion contours and rich textures, making it challenging to comprehensively perceive dense spatial motion. As a result, integrating the advantages of event cameras and traditional image cameras has become a new direction (Pan et al., 2020; Zhang et al., 2023). In this paper, we propose to adopt event data as

an auxiliary input to reconstruct point trajectories from input video and event sequences, modeling the comprehensive fine-grained spatial-temporal dense and continuous motion of the scene.

The introduction of event cameras offers the potential to model continuous motion from the data perspective. However, a new representation instead of optical flow is needed to parametrically associate the temporal dense properties of event data and model continuous long-term motion. BFlow (Gehrig et al., 2024) proposes to learn trajectories represented by Bézier curves from events, but is limited to motion between fixed frames and cannot adapt to longer sequences. CPFlow (Luo et al., 2023) proposes to learn control points represented as B-spline curves from a fixed number of image slices. Although continuous motion can be successfully modeled using these curve representations in normalized timescales, their fixed number of control points made it hard to handle complex dynamics and varying lengths of video sequences. Based on the curve representation, we propose a new streaming pipeline for accumulating multiple local curves to address these limitations.

In this paper, we present the first event-aided point tracking framework for recovering spatially dense and temporally continuous point trajectories from input videos and event sequences. Specifically, we first propose a new point trajectory representation with parametric curves that accumulate multiple local curves by learning offsets to adapt to multi-frame input videos at any length. We then design a new framework for combining two frames with events to simultaneously estimate dense point curve trajectories, and extend to multi-frame streaming. In addition, since most of the existing datasets lack continuous inter-frame motion annotations, we establish the association between continuous curve trajectory and event triggering as a part of the learning objective for continuous motion modeling. Extensive experiments on both simulated and real-world data demonstrate that the proposed framework significantly outperforms state-of-the-art methods. Particularly, our ablation studies illustrate the effectiveness of the proposed global aggregation and highlight the crucial role of incorporating event data in continuous trajectory modeling.

Our main contributions are summarized as follows:

- We introduce a new setup that, for the first time, enables long-term spatially dense and temporally continuous point tracking by integrating the strengths of both images and events.

- We present a novel global curve representation of continuous point trajectories through multi-frame aggregation, establishing a connection between event triggering and continuous motion.

- We propose a novel event-aided iterative streaming framework that accumulates the local tracks from two frames with inter-frame events, resulting in global, long-term dense and continuous trajectories through iterative temporal aggregation of global motion representation.

## 2 RELATED WORKS

### 2.1 IMAGE-BASED QUERY POINT TRACKING

The goal of point tracking is to recover the corresponding positions of query points in each frame, which has attracted wide attention with the proposal of the TAP benchmark and the baseline model TAPNet (Doersch et al., 2022). PIPs (Harley et al., 2022) proposes to extract independent point representation for 8-frame tracking through occlusion handling, then PIPs++ (Zheng et al., 2023) extends to long-term trajectories and Sun et al. (2024) extend to self-supervised refinement. TAPIR (Doersch et al., 2023) proposes a two-stage matching framework that fuses TAPNet and PIPs, and then BootsTAPIR (Doersch et al., 2024) adds self-supervised training on real data to improve robustness. Unlike these methods that track only one query at a time, CoTracker (Karaev et al., 2023) and Context-PIPs (Weikang et al., 2023) use additional tracks and pixel features as context information to improve global tracking performance. SpatialTracker (Xiao et al., 2024) introduces the triplane representation with depth prior to group pixels in 3D space. DINOTracker (Tumanyan et al., 2024) combines test-time self-supervised training based on the powerful pre-trained DINO-ViT (Oquab et al., 2024) model to achieve fine-grained tracking of a single video. When applying the above methods of tracking from query points to achieve dense tracking, points need to be processed individually or in batches, which brings computational hurdles and limits their downstream applications (Moing et al., 2024). Therefore, the current trend in point tracking tasks is to track dense points across the entire image in a single run, aiming to enhance neighborhood relationships while reducing computational requirements.

## 2.2 IMAGE-BASED DENSE POINT TRACKING

Recent studies turn to tracking every point within a frame simultaneously. OmniMotion (Wang et al., 2023) performs pixel-wise tracking via bijections between local and canonical space to maintain the global consistency of the motion, and then FastOmniTrack (Song et al., 2024) and DecoMotion (Li & Liu, 2024) improves from the perspectives of computational efficiency and object motion decomposition. CPFlow (Luo et al., 2023) proposes to estimate spatio-temporally dense motion curves, but it can only input 4 images and needs pre-sampling when inputting long-duration videos. AccFlow (Wu et al., 2023) proposes the forward and backward aggregation pipeline, extending interframe dense optical flow to multi-frame long ranges. MFT (Neoral et al., 2024) select chaining multi-frame candidates and FlowTrack (Cho et al., 2024) automatically apply error compensation in instances of tracking inaccuracies. DOT (Moing et al., 2024) unifies point tracking and optical flow, upgrading a small set of tracks to a dense flow field between arbitrary frames in a video. However, these methods are limited by the frame rate bottleneck of the input video and struggle to accurately model challenging dynamics. In this work, we propose to introduce continuous event data into the input video for spatially dense and temporally continuous point tracking.

## 2.3 EVENT-BASED MOTION ESTIMATION

Thanks to the motion-sensitive nature of event cameras, extensive motion estimation studies in recent years have highlighted their potential applications to challenging dynamics. The feature tracking methods (Gehrig et al., 2020b; Messikommer et al., 2023; Li et al., 2024; Wang et al., 2024) show the benefits of event cameras for low-latency tracking, but can only track sparse, specific textured locations. Recently estimating optical flow from events has become mainstream. Using only sparse event data (Zhu et al., 2019; Gehrig et al., 2021b; Luo et al., 2024) allows to estimate satisfactory dense optical flow, while introducing data from other sensors such as images (Wan et al., 2022; Zhou et al., 2024a) and point clouds (Wan et al., 2023; Zhou et al., 2024b) achieves significant performance gains. BFlow (Gehrig et al., 2024) and MotionPriorCMax (Hamann et al., 2024) exploits the continuous property of event data to estimate parametric Bézier trajectories, but can only estimate motion within a fixed consecutive frame interval and cannot be directly adapted to long-term sequences. Recently, FE-TAP (Liu et al., 2024) proposes to recover high-frame-rate point tracking from a fixed number of images and events based on TAPVid (Doersch et al., 2022), but does not take full advantage of the continuous nature of events for continuous trajectory modeling. We propose to combine the advantages of images and events to enable temporally continuous point tracking by modeling long-term global motion with an arbitrary number of frames.

## 3 METHOD

**Overview.** To the best of our knowledge, we present a first framework that recovers dense and continuous point trajectories from a video with corresponding event sequences. Our framework consists of four parts: 1) A parametric multi-frame continuous point trajectories representation; 2) An event triggered along the point trajectories model; 3) A two-frame basis motion estimation model; 4) A multi-frame motion aggregation and streaming framework.

**Problem Formulation.** A conventional shutter camera captures a video with $N_v$ frames of images $\{I_i\}_{i=1}^{N_v}$ at a fixed frame rate. An event camera generates an unbounded event sequence $\{e_i\}_{i=1}^{N_e}$ with independent pixels, where $N_e$ is the number of events. Each event $e_i = \{\mathbf{x}_i, t_i, p_i\}$ consists of the pixel position $\mathbf{x}_i = (x, y)$, timestamp $t_i$ with microsecond precision, and the brightness change polarity $p_i$ in logarithmic domain. Our goal is to combine the video and events to recover the spatially dense and temporally continuous trajectories $\mathbf{T}_{1 \to N_v}$ of all points starting from any instance of the first frame.

### 3.1 MOTION MODEL

**Trajectory representation.** Previous point tracking methods typically estimate some two-channel motion vectors in the xy directions, which is the optical flow when representing dense point trajectories (Cho et al., 2024; Moing et al., 2024). To learn the curve trajectory from the deep network, we instead learn the multiple control points of the curve (Luo et al., 2023). Specifically, we choose the

B-spline curve as our curve representation, which is defined by $N_c$ control points $\{\mathbf{P}_i\}_{i=1}^{N_c}$ and basis functions $\{B_{i,p}(t)\}_{i=1}^{N_c}$ with degree $p$. The continuous point trajectory $\mathbf{T}(t)$ represented by b-spline curve in time variable $t$ is a collection of piecewise polynomial functions $\mathbf{T}(t) = \sum_{i=1}^{N_c} B_{i,p}(t)\mathbf{P}_i$. More details are provided in the appendix. Similar to optical flow, each pixel has an independent curve with estimated control points denoted as $\mathbb{P} \in \mathbb{R}^{N_c \times 2 \times H \times W}$, where $H \times W$ is the image size. This realizes the learnable motion modeling purpose of dense and continuous point trajectories $\mathbf{T}$ with parametric curve representation.

**Multi-frame global trajectories accumulation.** Existing parametric motion modeling methods are fixed in the number of frames they can handle, *e.g.*, BFlow (Gehrig et al., 2024) is limited to between two frames, and CPFlow (Luo et al., 2023) struggles to benefit from more than 4 frame inputs, resulting in suboptimal long-term trajectory modeling. Inspired by the practice of multi-frame optical flow accumulation (Wu et al., 2023; Neoral et al., 2024), we propose a new multi-frame curve trajectories accumulation strategy to handle long-term videos with arbitrary frames.

Our accumulation framework works on a streaming pipeline, where the previous global trajectory $\mathbf{T}_{1 \to t}$ with $(t - 1) \times N_c$ control points has been accumulated from the previous $t - 1$ local trajectories $\{\mathbf{T}_{i \to i+1}\}_{i=1}^{t-1}$ when processing the $t$-th step. For an estimated $t$-th local trajectory as $\mathbf{T}_{t \to t+1}$ with $N_c$ control points from time $t$ to $t + 1$, a simple approach is to directly accumulate the initial current global trajectory $\mathbf{T}_{1 \to t+1}^{init}$ with $t \times N_c$ control points from time 1 to $t + 1$ by $\mathbf{T}_{1 \to t+1}^{init}(\mathbf{x}) = \left[\mathbf{T}_{1 \to t}(\mathbf{x}), \mathrm{Warp}\left(\mathbf{T}_{t \to t+1}, \mathbf{T}_{1 \to t}\right)(\mathbf{x})\right]$. $[,]$ combines the control points of two sub-curves and creates a more complex curve. However, there are two problems for the backward warping operation Warp: 1) It suffers from numerical error as integer sampling with floating-point coordinates is required, *i.e.*, for warping vectors from $\mathbf{b}$ to $\mathbf{a}$, $\mathrm{Warp}(\mathbf{a}, \mathbf{b})(\mathbf{x}) = \mathbf{a}(\mathbf{x} + \mathbf{b}(\mathbf{x}))$, $\mathbf{x}$ is integer coordinates but not $\mathbf{x} + \mathbf{b}(\mathbf{x})$. 2) Some points may be occluded at time $t$, resulting in failing to find the corresponding points.

Our framework iteratively maintains and learns to integrate from a global motion representation $\mathbf{M}_{1 \to t}^{global}$ in the streaming process. For the first numerical problem, we estimate a start point offsets $\mathbf{O}_t \in \mathbb{R}^{2 \times H \times W}$ learned from $\mathbf{M}_{1 \to t}^{global}$ and normalized to the range $[-1, 1]$. Sampling compensation is achieved by adding this offset directly during warping. For the second occlusion problem, we introduce an occlusion solving strategy for occluded pixels. We additionally estimate the visibility map $\mathbf{V}_{1 \to t}$ of each point from the initial frame to the $t$-th frame as well as the trajectory updates $\Delta \mathbf{T}_t$. Aggregation is based on a warp with offset when the point $x$ is visible. When point $x$ is occluded, a learnable module Fusion is introduced to regress the point's coarse motion trajectory in $t-> t+1$ from $\mathbf{M}_{1 \to t}^{global}$. Finally, the trajectory updates $\Delta \mathbf{T}_t$ are used to uniformly refine the final fine global trajectory. Our aggregation process can be modeled as follows:

$$\mathbf{T}_{1 \to t+1}(\mathbf{x}) = \begin{cases} \left[\mathbf{T}_{1 \to t}(\mathbf{x}), \mathrm{Warp}\left(\mathbf{T}_{t \to t+1}, \mathbf{T}_{1 \to t}, \mathbf{O}_t\right)(\mathbf{x}) + \Delta \mathbf{T}_t\right] & \text{if } \mathbf{V}_{1 \to t}(\mathbf{x}) = 1, \\ \left[\mathbf{T}_{1 \to t}(\mathbf{x}), \mathrm{Fusion}(\mathbf{T}_{t \to t+1}, \mathbf{T}_{1 \to t}, \mathbf{M}_{1 \to t}^{global})(\mathbf{x}) + \Delta \mathbf{T}_t\right] & \text{if } \mathbf{V}_{1 \to t}(\mathbf{x}) = 0. \end{cases} \quad (1)$$

**Events along the trajectory.** Following the contrast maximization framework (Gallego et al., 2018), we assume that events are triggered along with the pixel motion trajectories at the moving boundary. For a motion trajectory $\mathbf{T}$ starting from pixel $\mathbf{x}_1$ at time $t_1$, the generated events generated event's coordinates satisfy the trajectory, *i.e.*, $\mathbf{x}_1 = \mathbf{T}(t_1), \mathbf{x}_i - \mathbf{x}_1 = \mathbf{T}(t_i) - \mathbf{T}(t_1)$. We can thus use the motion trajectory to transform the following events back to time $t_1$:

$$e_i \doteq \{\mathbf{x}_i, t_i, p_i\} \to e_i' \doteq \{\mathbf{x}_i' = \mathrm{Warp}(\mathbf{x}_i; \mathbf{T}(t_i) - \mathbf{T}(t_1)), t_1, p_i\}. \quad (2)$$

Assuming the trajectory $\mathbf{T}$ is accurate, this process transforms the event $e_i$ to the starting point position $\mathbf{x}_1$ of the trajectory, *i.e.*, $\mathbf{x}_1 = \mathrm{Warp}(\mathbf{x_i}; \mathbf{T}(t_i - t_1))$. Based on the correlated motion modeling of events and point trajectories, we build additional self-supervised training objectives in Sec. 3.3 to alleviate the lack of continuous trajectory annotations in the training datasets.

### 3.2 FRAMEWORK

**Two-frame basis model.** The two-frame basis model is designed to recover inter-frame short-term trajectories $\mathbf{T}_{t \to t+1}$ from the encoded features of input two consecutive frames $F_t, F_{t+1}$ and

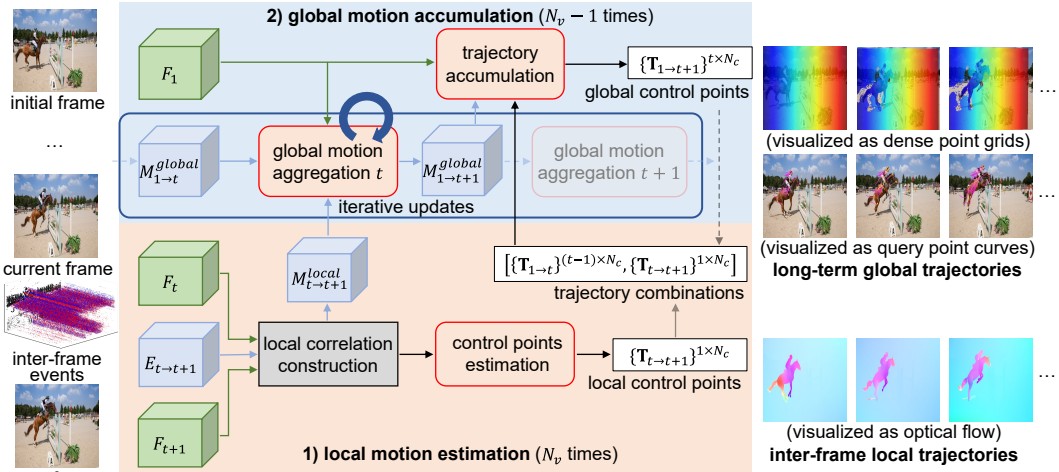

Figure 1: Our proposed event-aided dense and continuous point tracking framework consists of two main steps. 1) *Local motion estimation*: estimating short-term curve trajectories with $N_c$ control points from two consecutive images and inter-frame events, while concurrently updating the local motion representation. 2) *Global motion accumulation*: iteratively fusing the latest local motion representation with the previous global motion representation in a streaming manner for aggregating the latest global motion representation. Subsequently, the global long-term curve trajectories with $t \times N_c$ control points are optimized on trajectory combinations.

inter-frame events $E_{t \to t+1}$. This process involves three key components: feature extraction, local correlation construction, and control points estimation. In the feature extraction phase, we first convert the raw event data into a dense grid representation (Rebecq et al., 2019), followed by the feature encoding of the two-frame images and event grid, respectively. Subsequently, we construct the initial correlations between two frame features by matrix multiplication (Teed & Deng, 2020), and augment them with event features. By leveraging the local correlations and events, we learn the local motion representation $M_{t \to t+1}^{local}$ by a motion extractor which allows recovery of the dense trajectories $\mathbf{T}_{t \to t+1}$ by a trajectory decoder. Specifically, the trajectory decoder estimates the coordinates of $N_c$ control points $\mathbb{P}_{t \to t+1}$ and a single-channel visibility map $\{\mathbf{V}\}_{t \to t+1}$, which essential for establishing multi-frame global trajectories accumulation in Eq. 1.

**Global motion aggregation module.** In the context of processing a video comprising $N_v$ frames, the two-frame basis model described above needs to be streamed sequentially $N_v - 1$ times yielding local motion representations and local curve trajectories. To facilitate the accumulation of global multi-frame trajectories according to Sec. 3.1, the established global motion representation $\mathbf{h}_{t-1} \doteq \mathbf{M}_{1 \to t}^{global}$ from the previous $t$-frames is utilized as the query, while the current local motion representation $\mathbf{h}_t^l \doteq \mathbf{M}_{t \to t+1}^{local}$ serves as the key and value. We first perform the linear projections and compute the cross-attention:

$$\text{CA}(\mathbf{h}_{t-1}, \mathbf{h}_t^l, W_{Q,K,V}) = \text{softmax}\left(\frac{QK^T}{\sqrt{d_k}}\right)V = \text{softmax}\left(\frac{(W_Q \cdot \mathbf{h}_{t-1})(W_K \cdot \mathbf{h}_t^l)^T}{\sqrt{d_k}}\right)(W_V \cdot \mathbf{h}_t^l), \quad (3)$$

where $d_k$ is the channel size, $\cdot$ is the linear projection and $W_{Q,K,V}$ are the corresponding weights. We then conduct iterative fusion based on the gated activation unit (GRU) (Cho et al., 2014), where the update gate is $\mathbf{z}_t = \text{sigmoid}(\text{CA}(\mathbf{h}_{t-1}, \mathbf{h}_t^l, W_{Q,K,V}))$, the reset gate is $\mathbf{r}_t = \text{sigmoid}(\text{CA}(\mathbf{h}_{t-1}, \mathbf{h}_t^l, W'_{Q,K,V}))$, and the hidden state is $\mathbf{s}_t = \tanh(\text{CA}(\mathbf{r}_t \odot \mathbf{h}_{t-1}, \mathbf{h}_t^l, W''_{Q,K,V}))$, $\odot$ is the element-wise multiplication. The superscript of $W_{Q,K,V}$ denotes the different projection weights taken independently in each attention calculation. Finally, we iteratively update the current global motion representation in the feature level by:

$$\mathbf{M}_{1 \to t+1}^{global} \doteq \mathbf{h}_t = (1 - \mathbf{z}_t) \odot \mathbf{h}_{t-1} + \mathbf{z}_t \odot \mathbf{s}_t. \quad (4)$$

The simple and effective temporal aggregation we take is naturally compatible with the streaming pipeline, and also verifies its effectiveness in ablation experiments compared to previous solutions.

**Multi-frame iterative streaming framework.** As depicted in Fig. 1, our framework iteratively processes the input video and event data through local motion estimation and global motion accumulation. We aggregate the local motion representations from each frame interval to the global motion representation at the feature level through the above global aggregation module in Sec. 3.2. Subsequently, the multi-frame trajectory accumulation step described in Sec. 3.1 sequentially combines each inter-frame short-term curve into a global long-term motion trajectory at the trajectory level, providing the dense and continuous point tracking representation as the model output. On the right side of Fig. 1, local motion is visualized with dense optical flow, and global continuous motion is represented with deformations of dense point grid and curve trajectories of sparse query points.

### 3.3 Objective

**Temporal discrete trajectory supervision.** The available point tracking datasets provide only temporally discrete point tracks with no ground truth for continuous inter-frame trajectories. Following DOT (Moing et al., 2024), we first adopt supervised losses based on the temporal discrete ground-truth point tracks provided by the dataset, which consists of the L1 loss $L_{traj}$ for sampled discrete trajectory prediction and the binary cross-entropy loss $L_{vis}$ for visibility map.

We then randomly select different frame intervals for augmented training. Local correlation is not constructed when the frames are skipped, therefore the corresponding event features are taken into streaming for iteratively updating the global motion representation. There are cases where some images are not used as input when the frame interval is greater than 1, but the corresponding input events and ground-truth tracks can be regarded as inter-frame motion contributing to curve trajectory learning. Such sampling-based augmented training ensures the model learning through diverse long- and short-term motions, capitalizing on the continuity of events to estimate continuous trajectories.

**Event consistency with continuous trajectory.** Since events are usually generated along motion trajectories, we propose to leverage the continuous property of events for self-supervised continuous trajectory learning in conjunction with discrete supervision of point trajectories. However, events are computationally intensive to process one by one and are generally accompanied by noise. We thus first introduce event temporal chunking to process events in batches within a fixed duration to reduce the noise impact and computation. For the $b$-th interval of $B$ chunks, we isolate the events within that $b$-th chunk and aggregate them after warping them to $t_b$ as Eq. 2. For each chunk, the events then are summed into an image of warped events (IWE) (Gallego et al., 2018), *i.e.*, $\mathbf{EB}(\mathbf{x}_i, b) \doteq \sum_{i=1}^{N_e} \mathcal{N}(\mathbf{x}_i; x_i', \sigma^2)$, where $t_b \leq t_i < t_{b+1}$ and $\sigma$ is the neighboring range which is usually chosen as 1 pixel. This IWE essentially counts the number of warped events $e_i'$ per pixel and per chunk. The chunking intervals are chosen randomly to exploit the continuous nature of events. Thus we can establish consistent connections between event chunks and continuous trajectories:

$$L_{ec} = \sum_{\mathbf{x}}^{\Omega} \sum_{b_1 \neq b_2}^{B} \rho\Big(\mathbf{EB}(\mathbf{x}, b_1), \mathrm{Warp}\big(\mathbf{EB}(\mathbf{x}, b_2); \mathbf{T}(t_{b_2}) - \mathbf{T}(t_{b_1})\big)\Big), \quad (5)$$

where $\rho$ is the consistency measure by L1 norm. Since events are spatially sparse, we only establish connections with the dense trajectories at locations with valid events, which are denoted as $\Omega$.

**Image consistency with discrete trajectory.** Similar to the unsupervised optical flow task (Liu et al., 2020), we can also establish the discrete consistency of the motion trajectory with the images at discrete times, to compensate for the spatial sparsity issue of ground-truth point tracks in the training data. In addition, in our sampling-based augmented training, skipped images can be used as additional continuity training objectives.

For the accumulated continuous global trajectory $\mathbf{T}_{1 \to t}$, we sample the the discrete optical flow $\mathbf{F}_{i \to j}$ from $I_i$ to $I_j$ via timestamps. Similar to Eq. 5, the consistency of images can be modeled as:

$$L_{ic} = \sum_{\mathbf{x}} \sum_{i \neq j}^{N_v} \rho\Big(I_i(\mathbf{x}), \mathrm{Warp}\big(I_k(\mathbf{x}); \mathbf{F}_{i \to j}(\mathbf{x})\big)\Big). \quad (6)$$

**Total objective.** The total training objective is the weighted combination of the above objectives, *i.e.*, $L = L_{traj} + \lambda_1 L_{vis} + \lambda_2 L_{ec} + \lambda_3 L_{ic}$, $\lambda$ are manually hyperparameters. Our ablations verify that joint self-supervised training can compensate training for the temporal continuity of trajectories.

Table 1: Quantitative results of dense evaluation on the CVO test and extended set (Wu et al., 2023).

| | Method | CVO (Clean) | | CVO (Final) | | CVO (Extended) | |
|---|---|---|---|---|---|---|---|
| | | $EPE_{all/vis/occ} \downarrow$ | OA $\uparrow$ | $EPE_{all/vis/occ} \downarrow$ | OA $\uparrow$ | $EPE_{all/vis/occ} \downarrow$ | OA $\uparrow$ |
| Query | PIPs++ | 9.05 / 6.62 / 21.5 | 33.3 | 9.49 / 7.06 / 22.0 | 32.7 | 18.4 / 10.0 / 32.1 | 58.7 |
| | TAPIR | 3.80 / 1.49 / 14.7 | 73.5 | 4.19 / 1.86 / 15.3 | 72.4 | 19.8 / 4.74 / 42.5 | 68.4 |
| | CoTracker | 1.51 / 0.88 / 4.57 | 75.5 | 1.52 / 0.93 / 4.38 | 75.3 | 5.20 / 3.84 / 7.70 | 70.4 |
| Dense | GMA | 2.42 / 1.38 / 7.14 | 60.5 | 2.57 / 1.52 / 7.22 | 59.7 | 21.8 / 15.7 / 32.8 | 65.6 |
| | MFT | 2.91 / 1.39 / 9.93 | 19.4 | 3.16 / 1.56 / 10.3 | 19.5 | 21.4 / 9.20 / 41.8 | 37.6 |
| | AccFlow | 1.69 / 1.08 / 4.70 | 48.1 | 1.73 / 1.15 / 4.63 | 47.5 | 36.7 / 28.1 / 52.9 | 36.5 |
| | DOT | 1.32 / 0.74 / 4.12 | 80.4 | 1.38 / 0.82 / 4.10 | 80.2 | 5.07 / 3.67 / 7.34 | 71.0 |
| | EDCPT (Ours) | 1.23 / 0.71 / 3.83 | 82.1 | 1.31 / 0.76 / 3.86 | 81.9 | 4.88 / 3.44 / 7.46 | 71.9 |

## 4 EXPERIMENTS

### 4.1 EXPERIMENTAL DETAILS

**Datasets.** We follow the common evaluation practices in CoTracker (Karaev et al., 2023) and DOT (Moing et al., 2024). The training set MOVI-F (Greff et al., 2022) contains over 10,000 videos with 7 frames each. The CVO test (Wu et al., 2023) and extended (Moing et al., 2024) sets contain ∼500 videos with 7 and 48 frames respectively. The real test TAP-DAVIS benchmark (Doersch et al., 2022) includes 30 videos with ∼100 frames each. We simulate events for these two using the vid2e (Gehrig et al., 2020a) simulator. For the dense CVO dataset, we report the dense absolute error $EPE_{all/vis/occ}$ for all, visible and occluded points, as well as occlusion accuracy OA for estimated visible mask computed with IoU metric. For the sparse TAP-DAVIS dataset, we follow TAPNet (Doersch et al., 2022) by reporting average Jaccard AJ, position accuracy $<\delta^x_{avg}$, and occlusion accuracy OA. Additionally, we adopt the real-captured event-based optical flow dataset DSEC (Gehrig et al., 2021a;b) to verify the adaptation capacity, which contains 18 videos with ∼700 frames each.

**Implementation details.** We implement our model with PyTorch, train it on MOVI-F and directly evaluate it on CVO and TAP-DAVIS datasets. Following DOT (Moing et al., 2024), our model is trained for 500k steps on 4 × NVIDIA L40 48G GPUs, using the Adam optimizer and OneCycle learning rate decay with a maximum of $10^{-4}$. We also adopt the strategy of upgrading from multi-frame sparse to dense tracking in DOT to ensure temporal consistency. Unless specifically mentioned, we evaluate our models and competitors on the same PC with a single RTX 3090 GPU. We choose 3 frames as training samples, along with the random selection of up to 10 frames in different frame intervals. The loss hyperparameters are set to 1.0, 0.1, 0.1.

### 4.2 EXPERIMENTS WITH STANDARD SPATIALLY DENSE POINT TRACKING

**Performance on CVO and TAP-DAVIS benchmarks.** We first conduct a comprehensive evaluation of two commonly used datasets for point tracking tasks. Consistent with DOT (Moing et al., 2024), we report the quantitative results of the spatially dense optical flow from the last to the first frame of CVO (Wu et al., 2023) dataset in Table 1. Our proposed new EDCPT framework archives significant performance improvements, whether comparing methods that only predict the partial *Query* points or directly estimating spatially *Dense* trajectories within a single inference. Particularly, we achieve 0.19 $EPE_{all}$ and 0.9 OA improvements on the extended set of 476 videos with 48 frames when compared to the recent SOTA method DOT (Moing et al., 2024).

In contrast, the real TAP-DAVIS dataset (Doersch et al., 2022) only provides ground-truth trajectories for selected query points. As a result, we only sparsely evaluate these points for a fair comparison despite the output trajectories of our model and some compared methods are spatially dense. The quantitative results in Table 2 demonstrate the superiority of our framework, as evidenced by outperforming the existing state-of-the-art methods DOT (Moing et al., 2024) and SpatialTracker (Xiao et al., 2024) with up to 2.7 AJ and 1.1 OA. We also perform qualitative visual comparisons in Fig. 2 and in the Appendix. Combining the above quantitative and qualitative comparisons with previous image-based methods, the new attempts of incorporating events by our framework significantly improve the accuracy of standard dense point tracking tasks.

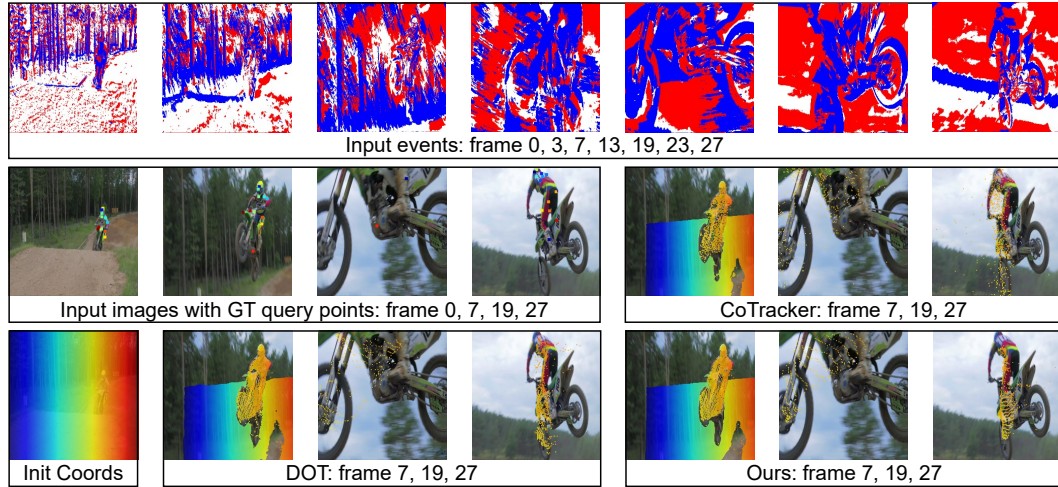

Figure 2: Visual comparisons of long-term dense point tracking on the *motocross-jump* sequence of TAP-DAVIS (Doersch et al., 2022), with the ground-truth sparse query points of input images.

Table 2: Quantitative results on the TAP-DAVIS (Doersch et al., 2022) point tracking benchmark.

| | Method | Source | DAVIS (First) | | | DAVIS (Strided) | | |
|---|---|---|---|---|---|---|---|---|
| | | | AJ ↑ | $<\delta^x_{avg}$ ↑ | OA ↑ | AJ ↑ | $<\delta^x_{avg}$ ↑ | OA ↑ |
| Query | TAP-Net | NeurIPS'22 | 33.0 | 48.6 | 78.8 | 38.4 | 53.1 | 82.3 |
| | Context-PIPs | NeurIPS'23 | 42.7 | 60.3 | 79.5 | 48.9 | 64.0 | 83.4 |
| | TAPIR | ICCV'23 | 56.2 | 70.0 | 86.5 | 61.3 | 73.6 | 88.8 |
| | CoTracker | arXiv'23 | 61.1 | 74.6 | 89.1 | 63.5 | 79.8 | 87.8 |
| | SpatialTracker | CVPR'24 | 61.1 | **76.3** | 89.5 | - | - | - |
| Dense | CPFlow | NeurIPS'23 | 9.6 | 14.6 | - | - | - | - |
| | MFT | WACV'24 | 47.3 | 66.8 | 77.8 | 56.1 | 70.8 | 86.9 |
| | DecoMotion | ECCV'24 | 53.0 | 69.9 | 84.2 | 60.2 | 74.4 | 87.2 |
| | DinoTracker | ECCV'24 | - | - | - | 62.3 | 78.2 | 87.5 |
| | FlowTrack | CVPR'24 | - | - | - | 63.2 | 76.3 | 89.2 |
| | DOT | CVPR'24 | 61.6 | 75.5 | 89.5 | 66.7 | **80.6** | 90.4 |
| | EDCPT (Ours) | - | **63.8** | **76.3** | **90.6** | **67.5** | 80.5 | **91.1** |

Table 3: Quantitative results on the DSEC optical flow leaderboard (Gehrig et al., 2021a). SSL denotes self-supervised learning and SL denotes supervised learning.

| Type | Method | Input | Source | EPE ↓ | AE ↓ | %Out ↓ |
|---|---|---|---|---|---|---|
| SSL | EV-FlowNet (Zhu et al., 2019) | Events | CVPR'19 | 3.86 | - | 31.45 |
| | Taming (Paredes-Vallés et al., 2023) | Events | ICCV'23 | 2.33 | 10.56 | 17.77 |
| | MPCMax (Hamann et al., 2024) | Events | ECCV'24 | 3.20 | 8.53 | 15.21 |
| SL | E-RAFT (Gehrig et al., 2021b) | Events | 3DV'21 | 0.79 | 2.85 | 2.68 |
| | TMA (Liu et al., 2023) | Events | ICCV'23 | 0.74 | 2.68 | 2.30 |
| | IDNet (Wu et al., 2024) | Events | ICRA'24 | 0.72 | 2.72 | 2.04 |
| | BFlow (Gehrig et al., 2024) | Events | TPAMI'24 | 0.75 | 2.68 | 2.44 |
| | BFlow (Gehrig et al., 2024) | Images + Events | TPAMI'24 | 0.69 | 2.42 | 1.88 |
| | EDCPT (Ours) | Images + Events | - | **0.64** | **2.17** | **1.64** |

**Performance on DSEC benchmark.** We further conduct experiments on the DSEC benchmark (Gehrig et al., 2021b) with real captured event data. Unlike the long-term global tracking goal of the point tracking task, the DSEC online leaderboard[1] only measures the optical flow between two consecutive frames. We therefore finetune the local motion estimation on the DSEC training

---

[1]https://dsec.ifi.uzh.ch/uzh/dsec-flow-optical-flow-benchmark

Table 4: Continuous point tracking evaluation results on the CVO extended set (Moing et al., 2024) and TAP-DAVIS dataset (Doersch et al., 2022).

| Method | CVO (Final) – $EPE_{all/vis/occ}$ ↓ | | | DAVIS (First) – AJ / $_{<\delta^x_{avg}}$ ↑ | | |
|---|---|---|---|---|---|---|
| | full | half | third | full | half | quarter |
| RAFT | 2.09 / 0.81 / 8.02 | 2.44 / 0.95 / 9.07 | 3.02 / 1.16 / 11.42 | 33.9 / 46.6 | 28.6 / 40.1 | 22.4 / 34.2 |
| GMA | 1.99 / 0.77 / 7.57 | 2.35 / 0.89 / 8.45 | 2.92 / 1.09 / 10.97 | 39.3 / 52.5 | 31.7 / 44.3 | 26.5 / 38.3 |
| AccFlow∗ | 2.28 / 0.60 / 11.18 | 2.39 / 0.80 / 10.74 | 2.79 / 1.02 / 11.37 | 47.2 / 62.3 | 37.5 / 49.2 | 30.9 / 42.4 |
| CoTracker | 1.89 / 0.63 / 7.05 | 2.11 / 0.82 / 8.02 | 3.17 / 1.65 / 11.13 | 61.1 / 74.6 | 54.3 / 68.8 | 48.9 / 63.9 |
| DOT | 1.83 / 0.59 / 6.95 | 2.10 / 0.73 / 7.88 | 2.69 / 0.97 / 10.84 | 61.6 / 75.5 | 55.6 / 70.1 | 50.4 / 65.3 |
| EDCPT (Ours) | **1.76 / 0.55 / 6.73** | **1.97 / 0.66 / 7.61** | **2.16 / 0.73 / 8.76** | **63.8 / 76.3** | **59.7 / 73.1** | **56.2 / 70.9** |

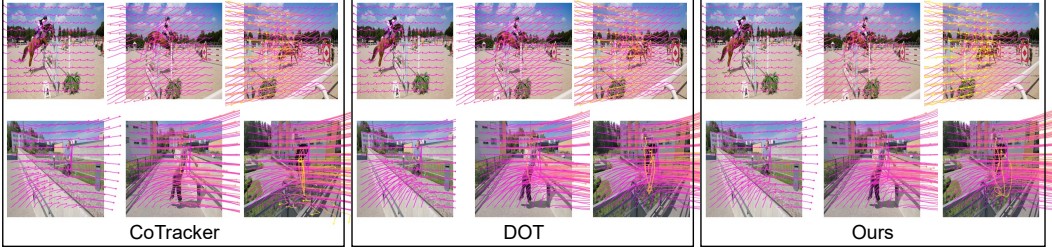

Figure 3: Visual comparisons of dense and continuous point trajectories on *horsejump-high* and *parkour* sequences of TAP-DAVIS (Doersch et al., 2022). Zoom in for detailed curve trajectories.

set from the pre-trained full model, and the submission results are shown in Table 3. Notably, while BFlow (Gehrig et al., 2024) can estimate inter-frame curve trajectories, they only submitted the optical flow version to the leaderboard. Our framework fuses images and events as well, yielding **1st rank** with performance improvements of 0.05 endpoint error (EPE) and 0.25 angular error (AE).

### 4.3 EXPERIMENTS WITH TEMPORALLY CONTINUOUS POINT TRACKING

We adapt the above standard procedure to input only a portion of the full video frames into the model to evaluate the temporal continuity with the ground truths of skipped frames. For the CVO final set with 7 frames per video, we skip 1-frame (*half*) and 2-frames (one-*third*) as model inputs, because the longer extended set lacks multi-frame ground-truth tracks. For the DAVIS dataset with ~100 frames, we report *half* at 1-frame and *quarter* at 3-frame intervals. The compared image-based methods lack the ability to model inter-frame motion, thus we take linear motion interpolation to generate trajectory when frames are skipped. We retrain AccFlow (Wu et al., 2023) and marked with ∗ as its public version for backward motion estimation does not support forward point tracking.

As reported in Table 4, our proposed new framework with global continuous trajectory accumulation significantly outperforms existing methods. Especially in nonlinear motion scenarios of DAVIS datasets, the larger frame intervals lead to greater performance gaps. In addition to the ablation of different motion assumptions in Table 6, the B-spline representation we adopt achieves better performance. We also provide a demo video[2] of continuous trajectory visualization in the Appendix, that includes visual comparisons of four sequences of simulated events from TAP-DAVIS and real captured events from ERF-X170FPS (Kim et al., 2023). Together with Fig. 3, we fully validate the capability of the proposed global motion accumulation in modeling continuous complex trajectories.

### 4.4 ABLATION EXPERIMENTS AND DISCUSSIONS

To perform progressive ablations in Table 5, 6, and 7, the underlined components are those utilized in the previous table, and the bolded ones represent the choices for our final framework. To validate the capability for continuous point tracking, the metrics for the ablation experiments are reported on the CVO third and DAVIS quarter settings in Sec. 4.3 and Table 4.

---

[2]Demo video: https://figshare.com/s/f96b1f1698adf2525fc0

Table 5: Ablations on global motion aggregation.

| | EPE$_{\text{all/vis/occ}}$ ↓ | AJ / $<\delta^x_{\text{avg}}$ ↑ |
|---|---|---|
| N/A | 2.54 / 0.89 / 10.33 | 51.9 / 66.4 |
| post | 2.49 / 0.85 / 9.73 | 52.5 / 66.9 |
| solo | 2.45 / 0.82 / 9.89 | 52.7 / 67.0 |
| −offsets | 2.43 / 0.82 / 9.73 | 53.0 / 67.2 |
| **stream** | **2.42 / 0.80 / 9.68** | **53.3 / 67.4** |

Table 6: Ablations on curve representation.

| | EPE$_{\text{all/vis/occ}}$ ↓ | AJ / $<\delta^x_{\text{avg}}$ ↑ |
|---|---|---|
| linear | 2.42 / 0.80 / 9.68 | 53.3 / 67.4 |
| quad | 2.49 / 0.84 / 9.46 | 54.2 / 68.1 |
| $N_c = 3$ | 2.32 / 0.79 / 9.33 | 54.4 / 68.6 |
| $\mathbf{N_c = 4}$ | **2.23 / 0.76 / 8.97** | **55.4 / 69.7** |
| $N_c = 5$ | 2.26 / **0.75** / 9.20 | 55.0 / 69.3 |

Table 7: Ablations on input data and supervision.

| | EPE$_{\text{all/vis/occ}}$ ↓ | AJ / $<\delta^x_{\text{avg}}$ ↑ |
|---|---|---|
| Images | 2.50 / 0.86 / 9.92 | 52.5 / 66.7 |
| +Events | 2.23 / 0.76 / 8.97 | 55.4 / 69.7 |
| +$L_{ic}$ | 2.20 / 0.75 / 8.90 | 55.8 / 70.2 |
| +$\mathbf{L_{ec}}$ | **2.16 / 0.73 / 8.76** | **56.2 / 70.9** |

**Global motion aggregation.** One of our key contributions is the global aggregation of local motion representations in the *stream*ing pipeline. Unlike direct process multi-frame optical flows (Wu et al., 2023), we aggregate motion representations at the feature level instead of dealing directly with motion vectors. Unlike temporal fusion with a fixed number of frames (Park et al., 2023), we adopt sequential modeling for temporal fusion with an unspecified number of frames in a streaming pipeline. In Table 5, N/A indicates that we do not explicitly model sequential motion as in DOT (Moing et al., 2024), *post* is the post-processing forward aggregation in AccFlow (Wu et al., 2023), and *solo* is the short and long term fusion module in SOLOFusion (Park et al., 2023). Our proposed motion aggregation framework, which fuses image correspondence and event features from local to global *stream*ing, achieves optimal performance. In addition, we also verified that removing the additional offset estimation to address the numerical problem in Warping leads to a slight performance degradation, as this would require the subsequent refinement to handle it simultaneously.

**Curve representation.** Based on the streaming aggregation framework, in Table 6 we compare the performance improvement of taking the B-spline curve representation compared to interpolating with *linear* and *quad*ratic motion assumptions. Based on experiments with different numbers of control points for B-spline curves, we chose $N_c = 4$, since further increasing the number of control points does not improve the performance under CVO third and DAVIS quarter settings.

**Input data and supervision.** Since previous methods usually use only image data, we evaluate the advantages of incorporating event data for high-precision continuous point tracking by removing events from our framework, as depicted in Table 7. Moreover, the comparison results between our image-only setting and DOT Moing et al. (2024) in Table 4 demonstrates that the proposed streaming aggregation and curve representation are beneficial even in the absence of event data. Furthermore, we validate that training using the proposed image and event-to-point trajectory consistencies as additional supervision complements the lack of continuous inter-frame tracks in the training data and can further improve performance.

**Limitations.** Our framework processes a 48-frame, 512x512 resolution video in 12.6 seconds, significantly faster than CoTracker, which takes 11 minutes while handling only partial query points in a single run. However, it is slower than the two-frame optical flow method GMA, which requires only 2.1 seconds, and slightly slower than the multi-frame method DOT, which takes 9.5 seconds. Due to the lack of real captured event-based point tracking datasets and challenges in obtaining long-term tracking labels, we evaluate point tracking on standard video benchmarks with simulated events and assess optical flow estimation quantitatively and point tracking qualitatively on real event datasets. Our future work plans to improve on model efficiency and evaluation dataset.

## 5 CONCLUSION

In this paper, we propose a new framework for integrating image and event data to estimate continuous motion trajectories for the emerging task of long-term dense point tracking. Specifically, we process the current two-frame images and inter-frame events in a streaming pipeline to estimate local motion representations, and combine previously established representations through global motion accumulation at the feature level to produce new global trajectories at the trajectory level. We utilize multi-frame parametric curve accumulation to represent continuous motion trajectories with any number of frames, complemented by image and event-to-trajectory consistency to enhance model training. We believe this work provides new insights into the point tracking task from the perspective of event-aided and continuous global curve representations.

ETHICS STATEMENT

This work is based entirely on publicly available datasets and does not involve any human subjects, animal experimentation, or sensitive data related to privacy or security. All datasets used in the experiments are open source available and do not contain any personal or confidential information. Therefore, there were no ethical issues associated with this study. The authors declare that they have no conflicts of interest.

REPRODUCIBILITY

Our experiments are conducted based on common code environments and hardware, using publicly available datasets. The details of the experimental setup are also described in the paper. To ensure reproducibility, our code will be publicly available.

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

# A   APPENDIX

In this appendix, we provide additional details of our methodology and experiments, the former including B-spline curve modeling and multi-frame trajectories aggregation, and the latter providing additional visualization results as well as a demo video on multiple datasets.

## A.1   METHOD DETAILS

**B-spline dense and continuous point trajectories.**   Given $N_c$ control points $\{\mathbf{P}_i\}^{N_c}$ and basis functions $\{B_{i,p}(t)\}^{N_c}$ with degree $p$, the continuous point trajectory $\mathbf{T}(t)$ represented by b-spline curve in time variable $t$ is a collection of piecewise polynomial functions:

$$\mathbf{T}(t) = \sum_{i=1}^{N_c} B_{i,p}(t)\mathbf{P}_i. \tag{7}$$

Based on the Cox–de Boor recursion, the detailed derivation of basis functions is:

$$B_{c,0}(t) = \begin{cases} 1 & k_i \leq t < k_{i+1} \\ 0 & \text{otherwise} \end{cases}, \tag{8}$$

$$B_{c,p}(t) = \frac{t - k_i}{k_{c+p} - k_i} B_{c,p-1}(t) + \frac{k_{c+p+1} - t}{k_{c+p+1} - k_{c+1}} B_{c+1,p-1}(t), \tag{9}$$

where $k_1, k_2, k_3, \ldots, k_m$ are $m = N_c + p + 1$ knots of the curve with a non-decreasing order that represent the times when the pieces polynomials meet. The internal $N_c - p + 1$ knots $k_{p+1}, k_{p+2}, \ldots, k_{m-p}$ constitute the deformation of the curve. The beginning and the ending remaining knots $k_1, k_2, \ldots, k_p$ and $k_{m-p+1}, k_{m-p+2}, \ldots, k_m$ are usually specified as duplicates of $k_{p+1}$ and $k_{m-p}$, in order to ensure the curve is tangent to the edges of the first and last control points so that the curve is clamped.

In experiments, we fixed the internal knots to evenly spaced numbers over a specified interval from 0 to 1, and the model only needs to learn the coordinates of control points $\{\mathbf{P}\}^{N_c} \in \mathbb{R}^{2 \times N_c \times H \times W}$ to model the continuous trajectory $\mathbf{T}$ of every pixel, where $H \times W$ is the image size. The head and tail of the modeled trajectory coincide with the start and end control points $\mathbf{P}_1$ and $\mathbf{P}_{N_c}$.

**Multi-frame optical flow and trajectories accumulation.**   Existing parametric motion modeling methods are fixed in the number of frames they can handle, *e.g.*, BFlow (Gehrig et al., 2024) is limited to between two frames, and CPFlow Luo et al. (2023) hard to get benefit for more than 4 frame inputs, resulting in suboptimal long-term trajectory modeling. Inspired by the practice of multi-frame optical flow aggregation Wu et al. (2023); Neoral et al. (2024), we propose a new multi-frame curve trajectories accumulation strategy to handle long-term videos with arbitrary frames.

In optical flow-based frameworks such as AccFlow (Wu et al., 2023) and MFT (Neoral et al., 2024), multi-frame optical flows are usually combined based on warping operations. Given the the previous global flow $\mathbf{F}_{1 \to t}$ and local flow $\mathbf{F}_{t \to t+1}$, representing the motion displacements from time 1 to $t$ and $t$ to $t + 1$, respectively, the aggregated current global flow $\mathbf{F}_{1 \to t+1} = [\mathbf{F}_{1 \to t} \oplus \mathbf{F}_{t \to t+1}]$ from time 1 to $t + 1$ can be computed as follows:

$$\mathbf{F}_{1 \to t+1}(\mathbf{x}) = \begin{cases} \mathbf{F}_{1 \to t}(\mathbf{x}) + \text{Warp}\left(\mathbf{F}_{t \to t+1}, \mathbf{F}_{1 \to t}\right)(\mathbf{x}) & \text{if } \mathbf{V}_{1 \to t}(\mathbf{x}) = 1, \\ \mathbf{F}_{1 \to t}(\mathbf{x}) + \text{Fusion}\left(\mathbf{F}_{t \to t+1}, \mathbf{F}_{1 \to t}\right)(\mathbf{x}) & \text{if } \mathbf{V}_{1 \to t}(\mathbf{x}) = 0, \end{cases} \tag{10}$$

where $\mathbf{V}_{1 \to t}(\mathbf{x})$ indicates whether the point $\mathbf{x}$ from time 1 is visible at time $t$. $[,]$ denotes the aggregation operation, Warp is the backward warping operation. Fusion is the additional occlusion solving by fusing the residual flow if pixels are occluded and cannot be directly aggregated. Notably, the warping operation has an inherent error as it requires integer sampling with floating-point coordinates, *i.e.*, $\text{Warp}(\mathbf{a}, \mathbf{b})(\mathbf{x}) = \mathbf{a}(\mathbf{x} + \mathbf{b}(\mathbf{x}))$. Therefore, an additional post-refinement is still necessary even in unoccluded areas.

In contrast, multi-frame curve aggregation also considers how to keep the shape of the subcurves while aggregating the curves. Denote the previous global curve as $\mathbf{T}_{1 \to t}$ with $(t - 1) \times N_c$ control points, which represents the aggregation of $t - 1$ sub-curves $\mathbf{T}_{1 \to 2}, ..., \mathbf{T}_{t-1 \to t}$ from time 1 to $t$.

If we get the local sub-curve piece as $\mathbf{T}_{t \to t+1}$ with $N_c$ control points from time $t$ to $t+1$, we can propagate the current global trajectory $\mathbf{T}_{1 \to t+1} = [\mathbf{T}_{1 \to t} \oplus \mathbf{T}_{t \to t+1}]$ with $t \times N_c$ control points from time 1 to $t+1$ by:

$$\mathbf{T}_{1 \to t+1}(\mathbf{x}) = \begin{cases} \text{Aggreg}\big(\mathbf{T}_{1 \to t}(\mathbf{x}), \text{Warp}\,(\mathbf{T}_{t \to t+1}, \mathbf{T}_{1 \to t})\,(\mathbf{x})\big) & \text{if } \mathbf{V}_{1 \to t}(\mathbf{x}) = 1, \\ \text{Aggreg}\big(\mathbf{T}_{1 \to t}(\mathbf{x}), \text{Fusion}\,(\mathbf{T}_{t \to t+1}, \mathbf{T}_{1 \to t})\,(\mathbf{x})\big) & \text{if } \mathbf{V}_{1 \to t}(\mathbf{x}) = 0, \end{cases} \quad (11)$$

where Aggreg aggregates the control points of two sub-curves to create a more complex smooth curve.

Taking two curves $\mathbf{T}_1$ and $\mathbf{T}_2$ with $N_1$ and $N_2$ control points $\{\mathbf{P}_i\}^{N_1}$ and $\{\mathbf{Q}_i\}^{N_2}$ respectively as an example, the aggregation process smoothly connects the two curves while ensuring the resulting curve goes through the endpoints of the sub-curves, *i.e.*, the first start point $\mathbf{P}_1$, the first endpoint $\mathbf{P}_{N_1}$ (overlapped with the second start point $\mathbf{Q}_1$), and the end point of $\mathbf{Q}_{N_2}$. To achieve this, we need to ensure that both the position, tangent and curvature (0th, 1st, 2nd order derivatives) are continuous at the position of the connected points, *i.e.*, $\mathbf{Q}'_1 = \mathbf{P}_{N_1}$, $\mathbf{Q}'_2 - \mathbf{Q}'_1 = s_1(\mathbf{P}_{N_1} - \mathbf{P}_{N_1-1})$, and $\mathbf{Q}'_3 - \mathbf{Q}'_1 = s_2(\mathbf{P}_{N_1} - 2\mathbf{P}_{N_1-1} + \mathbf{P}_{N_1-2})$, where $s_1, s_2$ are the scaling factors usually set to 1, $\mathbf{Q}'$ represent the updated control points of the second curve. This process is included in the Update operation along with the trajectory updates $\Delta T$ prediction. In addition, since the modeled curves usually end in floating-point coordinates but start at integers on the image grid, we need to take bilinear interpolation in neighborhoods $\delta$ to establish the aggregation, denoted as Interp. Altogether, the aggregation process can be expressed as:

$$\text{Aggreg}(\mathbf{T}_1, \mathbf{T}_2) = \text{Concat}\Big(\{\mathbf{P}_i\}^{N_1}, \text{Update}\big(\text{Interp}(\{\mathbf{Q}_i\}^{N_2})\big)\Big). \quad (12)$$

where the control points of the original first curve and the control points of the updated second curve are concatenated together to get $N_1 + N_2$ control points. Then the corresponding modifications get $N_1 + N_2 + p + 1$ knots, which gives the aggregated long-term global trajectory.

We simplify the expression of the above procedure in Eq. 1, *i.e.*, the Aggreg process corresponds to the combination operation $[,]$, and Update consists of the third-order alignment and residual $\Delta T$ update from two sub-curves to a global curve.

## A.2 EXPERIMENTAL DETAILS

**Qualitative visual comparisons.** Due to the length limitation, we provide more visualization results of point tracking in this appendix. Fig. 4 and Fig. 5 show the results on the TAP-DAVIS and CVO datasets, where we achieve better point tracking performance compared to recent competitive methods It is worth noting that the TAP-DAVIS dataset Doersch et al. (2022) only provides sparse query point trajectories for each frame, so we plot the positions of the ground-truth query points directly on the input image, while initial point coordinates (*Init Coords*) represent the initial coordinates of dense point tracking. In contrast, the CVO extended set Moing et al. (2024) has only the last frame of the dense point motion vectors, so we provide the visualization of the ground-truth points (*GT points*) from the Init Coords of the first frame to last frame.

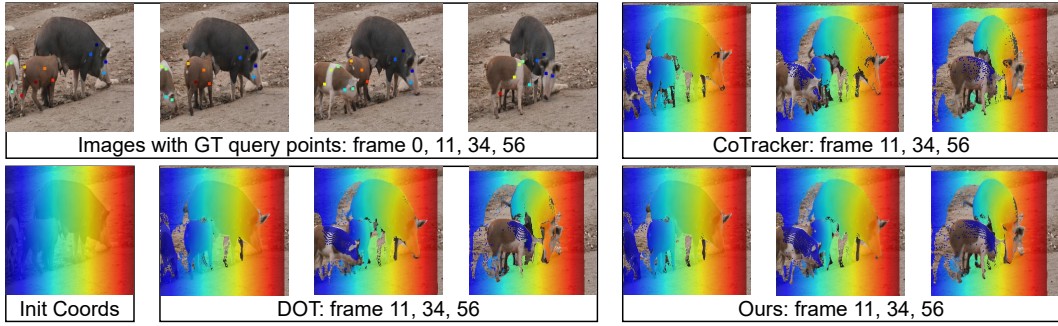

Figure 4: Visual comparisons of long-term dense point tracking on the *pigs* sequence of TAP-DAVIS (Doersch et al., 2022), with the ground-truth sparse query points of input images.

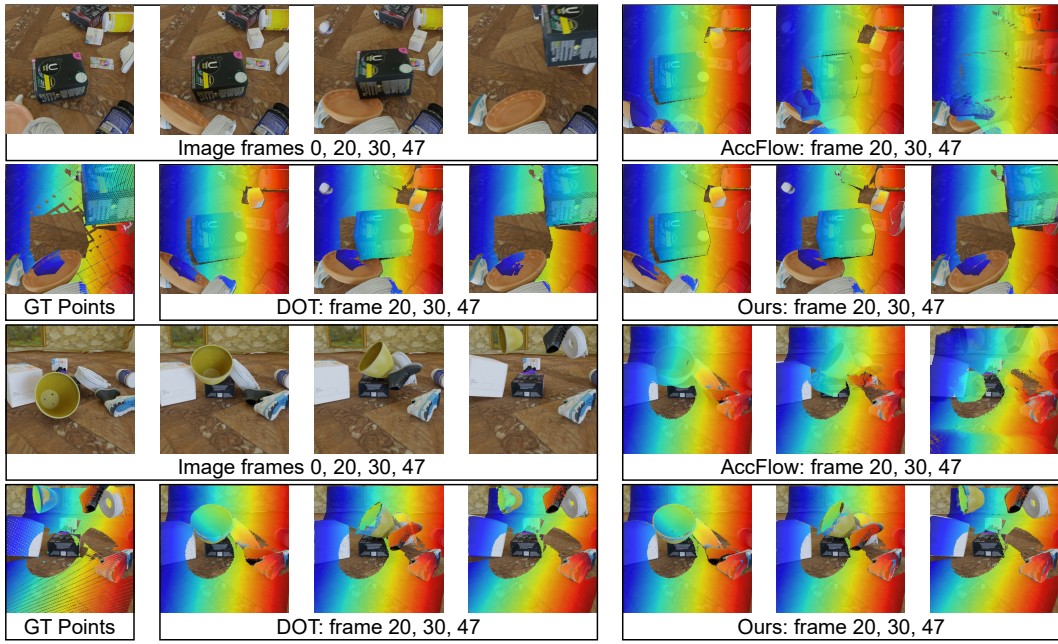

Figure 5: Visual comparisons of dense point tracking on the CVO extended set (Moing et al., 2024) with the ground-truth dense point coordinates at the last (48-th) frame.

| Rank ▲▼ | method ▲▼ | Details ▲▼ | 1PE ▲▼ | 2PE ▲▼ | 3PE ▲▼ | EPE ▲▼ | AE ▲▼ |
|---|---|---|---|---|---|---|---|
| 1 | EDCPT | | 7.05 | 2.481 | 1.637 | 0.636 | 2.166 |
| 2 | STFlow | | 8.58 | 2.928 | 1.677 | 0.663 | 2.369 |
| 3 | EFECM | | 8.378 | 2.868 | 1.696 | 0.668 | 2.524 |
| 4 | ECDDP | | 8.887 | 3.199 | 1.958 | 0.697 | 2.575 |
| 5 | IDNet | Details | 10.069 | 3.497 | 2.036 | 0.719 | 2.723 |
| 6 | TMA | | 10.863 | 3.972 | 2.301 | 0.743 | 2.684 |
| 7 | EEMFlow+ | | 11.403 | 3.932 | 2.145 | 0.751 | 2.669 |
| 8 | eventRanger | | 11.322 | 4.12 | 2.349 | 0.754 | 2.711 |
| 9 | E-Flowformer(BlinkFlow) | | 11.225 | 4.102 | 2.446 | 0.759 | 2.676 |
| 10 | ADMFlow | | 12.522 | 4.673 | 2.647 | 0.779 | 2.838 |
| 11 | E-RAFT | Details | 12.742 | 4.74 | 2.684 | 0.788 | 2.851 |
| 12 | E-RAFT* | | 16.193 | 6.22 | 3.594 | 0.901 | 3.126 |
| 13 | STTFlowNet | | 18.166 | 7.732 | 4.588 | 0.997 | 3.235 |
| 14 | SDformerFlow | Details | 37.576 | 17.123 | 10.051 | 1.602 | 4.871 |

Figure 6: Screenshot of the DSEC optical flow leaderboard (Gehrig et al., 2021a) on Sept. 30, 2024 from `https://dsec.ifi.uzh.ch/uzh/dsec-flow-optical-flow-benchmark`. Our proposed EDCPT achieves the current first rank in the DSEC optical flow benchmark.

**Experimental result on the DSEC benchmark.** To qualitatively validate the applicability of our scheme on real captured events data, we conduct experiments on DSEC (Gehrig et al., 2021a) , a widely used benchmark for optical flow estimation, and submit the results on the test set to DSEC online leaderboard. In Table 3, we compare the performance of various SOTA methods under dif-

ferent training and input settings, here we also provide a screenshot of the DSEC online leaderboard in Fig. 6. Our proposed EDCPT achieves the current first rank in the DSEC optical flow benchmark.

**Demo video.** The demo video is uploaded anonymously to `https://figshare.com/s/f96b1f1698adf2525fc0`. We recommend accessing the high-resolution version of video `1295_demo_video.mp4` from the Supplementary Material. In this appendix, we provide screenshots of the demo videos. Fig. 7 shows the video screenshots for the comparison results of dense and continuous point tracking in four scenes, including the *horsejump-high* and *parkou* sequences on the TAP-DAVIS dataset Doersch et al. (2022), and the *test_0005* and *test_0033* sequences on the real-captured ERF-X170FPS dataset Kim et al. (2023). We chose to compare with two recent SOTA methods, CoTracker Karaev et al. (2023) and DOT Moing et al. (2024). The visualization of dense and continuous point tracking trajectories is shown in three separate forms: query point trajectories, grid trajectories, and dense point coordinate shifts.

In particular, the ERF-X170FPS dataset is proposed in CBMNet Kim et al. (2023) originally for video frame interpolation in highly dynamic scenarios. Since both its image and event data are real-captured and of high quality, we utilize it to further validate the applicability of our framework on real-world data. Since this dataset lacks motion annotations and query point coordinates, we only show grid trajectories and dense point coordinate shifts. As shown in the demo video and screenshots in Fig. 7, our framework achieves better point tracking performance compared to Cotracker and DOT for small objects (soccer ball in *test_0005*) and curve motion (camera rotation in *test_0033*).

**TAP-DAVIS: horsejump-high**

**TAP-DAVIS: parkour**

**ERF-X170FPS: test_0005**

**ERF-X170FPS: test_0033**

Figure 7: Screenshots from our demo video, including comparisons of dense and continuous point tracking trajectories on the commonly used TAP-DAVIS benchmark (Doersch et al., 2022) and the real-world ERF-X170FPS dataset (Kim et al., 2023).

