# OpenReview forum: "Event-aided Dense and Continuous Point Tracking"
_ICLR.cc/2025/Conference — ICLR 2025 Conference Withdrawn Submission_

### Official Review · Reviewer_WR11 · 2024-10-28

**Soundness:** 2
**Presentation:** 3
**Contribution:** 2
**Rating:** 5
**Confidence:** 4

**Summary:**

This paper performs dense point tracking to estimate large motion in long video sequences using events. To solve this task effectively, a multi-frame iterative framework is proposed, which estimates inter-frame motion and uses an aggregation method to estimate global motion. This approach was evaluated on multiple datasets, highlighting its strengths and opening a new field in motion estimation for event cameras.

**Strengths:**

- The writing in this paper is very straightforward, making it easy to read even for those new to events or point tracking. The structure is well-composed, with sufficient references, which is commendable.

- Another strength is in the evaluation on multiple datasets—not only on synthetic ones but also on real-world event datasets. This aligns with the evaluation protocols of many prior studies, which adds credibility.

- Various supplementary materials and extensive experimental data also enhance the paper's quality.

**Weaknesses:**

- The novelty of the proposed method is difficult to discern. It appears to be a straightforward adaptation of prior point tracking methods with event stacking. For instance, compared to methods like FlowTrack or DOT, it’s challenging to see any distinctive or novel aspects. Specifically, the approach of estimating local motion and then accumulating it is an old technique, commonly used in optical flow and dense tracking.

- Another drawback is the lack of an inference time comparison, which is a common benchmark in prior protocols (e.g., in the FlowTrack paper). While comparing all methods on both synthetic and real datasets may be impractical, comparison with some representative studies is essential.

- Since there’s no dedicated event-based dense tracking dataset, the authors rely on synthetic datasets and evaluate event-based optical flow in real-world settings. However, this does not truly reflect event-based dense tracking, which is a significant weakness.

**Questions:**

- The author needs to emphasize the distinct aspects of the framework compared to existing methods beyond merely adding events. Currently, these differences are hard to identify.

- Demonstrating the framework’s effectiveness through computational cost analysis would support that it’s more than just a parameter-heavy approach and instead an efficient method.

- While real-world experiments would be ideal, it’s understandable that this may be infeasible within the given timeframe.

---

### Official Review · Reviewer_5NUL · 2024-11-03

**Soundness:** 2
**Presentation:** 2
**Contribution:** 2
**Rating:** 5
**Confidence:** 4

**Summary:**

The paper introduces a novel event-aided dense and continuous point tracking framework (EDCPT) that integrates the strengths of both image and event data to achieve high-resolution motion tracking in video sequences. The method proposes a multi-frame aggregation strategy for dense point tracking, leveraging event cameras to address temporal limitations in conventional video data. Through this approach, EDCPT can capture temporally continuous point trajectories, which is validated by experiments showing significant performance gains over existing state-of-the-art methods in dense tracking tasks.

**Strengths:**

- The method addresses a key limitation in point tracking by integrating event cameras, effectively combining their high temporal sensitivity with the spatial information of traditional video.
- The proposed multi-frame iterative streaming process for motion aggregation is well-designed and enables the model to adapt to variable video lengths.
- The paper provides comprehensive experiments on simulated and real-world datasets. The results convincingly demonstrate the advantages of using event data for fine-grained and continuous tracking, with significant improvements over baseline methods.

**Weaknesses:**

- The reliance on event data limits the framework's flexibility, as it may not perform optimally without event camera input. This restricts its applicability to setups where event cameras are available.
- Some technical assumptions are not fully supported by the results. For instance, while the multi-frame aggregation is shown to improve performance, there is limited analysis of its specific contribution compared to simpler aggregation techniques.
- The framework’s computational cost, especially given the use of multi-frame input and dense tracking, could make it challenging for use in real-time applications, which is not fully addressed in the paper.
- The EDCPT framework is computationally demanding, limiting its real-time applicability in scenarios where immediate results are required. Additionally, its reliance on event cameras restricts its use to specific hardware configurations, reducing its flexibility. While the proposed method is validated on benchmark datasets, further testing in a broader range of real-world applications would strengthen the claims of generalizability. Finally, the integration of event data introduces complexity in the framework, which may pose challenges in deployment and necessitate robust calibration and setup procedures for optimal performance.

**Questions:**

- How does the framework handle sparsity or noise in the event data? Since real-world event cameras often produce sparse or noisy data, it would be valuable to understand the robustness of the proposed method in these conditions.
- Some parts of the article are a bit vague and need more explanation. For example, the paper mentions an occlusion handling strategy but lacks quantitative evidence of its effectiveness. Could the authors provide more information on how occlusions are managed and evaluated?

---

### Official Review · Reviewer_zoGa · 2024-11-04

**Soundness:** 2
**Presentation:** 1
**Contribution:** 2
**Rating:** 3
**Confidence:** 4

**Summary:**

The paper proposes a model for dense point tracking from a multimodal input consisting of RGB frames captured by a standard shutter camera and events coming from an event camera.

In order to represent the motion, the method proposes to parametrize point trajectories with B-Splines, predicting therefore the control points {P_i}_i=1...Nc to recover the curves {T_t}.

Their proposed "local motion etimation" model operates on pairs of adjacent frames I_t and I_{t+1} as well as the events happening between these adjacent frames E_{t->t+1}; and predicts local trajectories {T_{t->t+1}} between these adjacent frames. Then, these trajectories are sequentially combined to obtain {T_{1->t}}, a process which is sometimes aided by the current global motion representation M^{global} in case of occlusions (eq. 1). This global motion representation M^{global} is iteratively updated using the local motion representations M^{local} extracted by the "local motion estimation" module.

The model is trained on the synthetic MOVI-F dataset with 10k 7-frame videos during 500k steps, and evaluated on CVO-test and TAPVid-DAVIS. For the evaluation datasets, events are simulated using vid2e.

Quantitative results show the proposed model can obtain SOTA performance on TAPVid-DAVIS and CVO point-tracking benchmarks, as well as in the DSEC optical flow leaderboard.

The authors also present ablation experiments for their global motion aggregation, curve representation and input data.

**Strengths:**

- While combining event data with RGB data had been used in previous work in optical flow, this is the first work using event data for long-range point tracking.

- The authors conduct experimental evaluations on standard point-tracking benchmarks and report SOTA results.

**Weaknesses:**

- The method is not fully understandable nor reproducible with the details given. Figure 1 gives the reader the best guess about how the method works, but it is not really clear how the events are processed by the model, how the local motion representations are obtained, what is the trajectory decoder (L247) and how the global representations are are used for the final trajectory predictions.

- It's not clear how the events are obtained for the synthetic MOVI-F training data.

- Overall the paper is poorly written and difficult to understand. There are errors that show it was not carefully proofread, there are organization issues, and there are notation issues. For example, sec 3.1 speaks about the global motion representation without having introduced it. The notation in sec. 3.1 is also difficult to follow. For instance, it's not clear what the "initial current global trajectory" T^init_{1->t+1} means and how it is used, as it doesn't appear in any equation. There are also no details about that the Warp and Fusion operations in eq (1) are.

**Questions:**

- Please explain how the events are used for the local correlation construction and how the M^{local} is computed.

- Please explain what are the warp and fusion operations in (1).

- Please explain how the global motion representations M^{global} are used.

- Please explain how events are obtained for the training data.

---

### Note · Authors · 2024-11-15

I have read and agree with the venue's withdrawal policy on behalf of myself and my co-authors.